# Sex as a moderator of the relationship between hip abduction strength and muscle activation during single-leg stance

Rodrigo Rodrigues[1,2,3], Francesca Chaida Sonda[3], Michele Fernandes Frigotto[4], Pablo Gaviraghi[4], Talita Molinari[2], Nicolas da Silva Pereira[2], Matheus Iglesias Marques[1,2], Rodrigo Freire Guimarães[1,2], Eduarda Bastos Cabral[1,2], Rodrigo Rabello[5]*

1 Graduate Program in Health Science, Federal University of Rio Grande, Rio Grande, Brazil, 2 Sports and Exercise Neuromechanics Group, Federal University of Rio Grande, Rio Grande, Brazil, 3 Exercise Research Laboratory, Federal University of Rio Grande do Sul, Porto Alegre, Brazil, 4 Exercise Physiology and Physical Assessment Laboratory, Serra Gaúcha University Center, Caxias do Sul, Brazil, 5 Sports and Exercise Medicine, Queen Mary University London, London, United Kingdom

* r.rabellodasilva@qmul.ac.uk

## Abstract

### Background

Single-leg stance requires pelvic stability, largely supported by the hip abductors. Differences in hip abductor activation between sexes and individuals with or without musculoskeletal conditions may relate to abductor weakness. However, the relationship between hip abduction strength and muscle activation during stance, and whether this is moderated by sex, remains unclear.

### Objective

To investigate whether maximal hip abduction strength is associated with hip abductor EMG amplitude during single-leg stance, considering sex as a moderator.

### Methods

Thirty-six adults (18 males, 18 females) performed an estimated 1RM side-lying hip abduction test and two 10-second single-leg stance trials. EMG amplitude of the gluteus medius (GMed) and tensor fasciae latae (TFL) was analyzed. A moderation analysis (PROCESS Model 1) was used to test the interaction between strength and sex.

### Results

Hip abduction strength, sex, and their interaction explained 51% of the variance in GMed EMG amplitude ($R^2 = 0.51$; $p < 0.001$). A significant strength × sex interaction were observed ($p = 0.002$). Females with lower strength showed greater GMed

**Data availability statement:** All data are in the paper and/or supporting information files.

**Funding:** The author(s) received no specific funding for this work.

**Competing interests:** The authors have declared that no competing interests exist.

activation ($p < 0.001$); this was not seen in males ($p = 0.24$). No significant effects were found for TFL activation ($R^2 = 0.02$; $p = 0.89$).

## Conclusion

Females with lower hip abduction strength demonstrate greater GMed activation during single-leg stance, suggesting a sex-specific compensatory strategy. No similar effect was observed for TFL. These findings highlight the importance of considering sex in neuromuscular assessments of pelvic stability.

## Introduction

During various motor tasks, such as gait or single-leg squat, single-leg stance positions are required [1]. In addition to other muscle synergies, this position involves a maintenance of pelvis frontal plane alignment, which is a specific role of gluteus medius (GMed) [2]. Thus, differences in the muscle activation of GMed have been reported between people with and without low back [3] and knee pain [4–6]. The primary reason for differences in EMG amplitude is attributed to hip abductor weakness [3,7,8]. This concept may support two contrasting ideas: (i) when the muscles are weak, their activation is also lower as they compensate this hip abductor weakness by increasing ipsilateral trunk lean [4,9]; (ii) when the muscles are weak, their activation is higher as a possible compensatory effect to maintain pelvis alignment [7,10]. Moreover, other compensatory strategy could be increasing the activation of synergists, such as the tensor fascia lata (TFL), an important hip abductor when the hip is extended [2]. In fact, it is possible that this is a cyclic mechanism whereby the muscles are initially weak due to this inability to activate the muscle across activities. If either of these ideas are true, we would expect to find an association between strength level and EMG amplitude during single-leg tasks.

Sex is a potential moderator, given that similar explanations have been reported in male–female comparisons [4,8]. Historically, females have been underrepresented in physiological studies, with findings from male samples often being generalized to females [11]. This is a significant issue, as there are several biological differences in females that are not considered and may provide advantages to females in sustained tasks [12]. Moreover, there is a growing body of research in exercise science that considers sex differences, driven by the increasing participation of females in various sports [13], but with higher rates of some injuries [3,4,14]. One emerging aspect is the understanding of how neuromuscular control strategies are achieved [15], which appear to differ between sexes [16,17] and may also be reflected in EMG amplitude. Our study aims to examine whether there is an association between maximal strength and EMG amplitude of the GMed and TFL during single-leg stance, with sex as a potential moderator of this relationship.

## Methods

### Participants

Males and females were recruited at university campus. Eligible participants had no history of lower limb injuries and participated in strength training programs for at least

three months at the time of data collection. The study was conducted in accordance with the principles of the Declaration of Helsinki, and all participants provided written consent after receiving a thorough explanation of the procedures involved. This study was approved by the university's ethics committee (n: 3.446.338).

## Procedures

All assessments were completed within a single visit, following a predetermined sequence: (i) evaluation of body composition parameters and training history; (ii) assessment of hip abductors Maximal Isometric Voluntary Contraction (MIVC); (iii) single-leg stance task; (iv) estimation of 1RM for the side-lying hip abduction exercise.

**Hip abductors maximal voluntary isometric contractions.** After 15 submaximal repetitions of side-lying hip abduction for warm-up, 3 MVIC were performed with the hip of the dominant side abducted at a 10° angle, while recording the EMG signal of the GMed and TFL. The contralateral leg was flexed at both the hip and knee to 90°. To ensure standardized resistance, the distal region of the dominant leg (approximately 5 cm above the lateral malleolus) was securely positioned against a rigid and fixed structure [8].

**Single-leg stance.** Participants performed two 10-second single-leg stance tasks with 30 seconds of rest between attempts [18]. The test was performed barefoot on a firm surface, with participants' arms crossed over their shoulders, eyes open, and the contralateral knee flexed. If the participant lost balance and touched the floor with the other foot, the attempt was considered invalid and had to be repeated. The mean of the two valid attempts was used for analysis.

**EMG data acquisition and analysis.** An electromyographer with a sampling rate of 2000 Hz and 14-bit resolution (Miotool-400, Miotec – Biomedical Equipment, Porto Alegre, Brazil) was used to capture the activation of the GMed and TFL during MVIC and single-leg stance tasks. After skin preparation, two electrodes with a radius of 15 mm (Kendall Mini MediTrace 100 – Tyco Healthcare, São Paulo, SP, Brazil) were placed 20 mm apart (center-to-center) on the skin over each muscle belly, following the guidelines outlined by SENIAM.

EMG data were filtered using a 4th order Butterworth filter (bandpass: 20–500 Hz), rectified and smoothed using a low-pass 6 Hz Butterworth filter. The highest RMS value of GMed and TFL obtained during the five-second period of MVIC was recorded as the maximal activation. During the single-leg stance task, the RMS average activation was computed and normalized by MVIC of the respective muscles (expressed as a percentage of maximal activation). All EMG data analysis were conducted using a custom-written MATLAB script (vR2021b, Mathworks Inc., Natwick, WY, United States).

**1RM test.** To estimate 1RM, the weight used was multiplied by the Lombardi coefficients according to the number of repetitions performed (up to a maximum of 10), and the resulting value was calculated. The test consisted of side-lying hip abduction exercises, initially performed with ankle weights corresponding to 20% of each participant's body weight. Execution speed was controlled using a metronome set at 60 beats per minute, with two beats allocated for each phase of the movement. If participants completed more than 10 repetitions with the initial load, a three-minute rest was provided, and the load was increased by 10% for a new attempt [17]. Both absolute load (expressed in kg) and relative load (normalized to body mass and expressed in kg/kg) were included in the analysis.

## Statistical analysis

The assumptions of linear regression were first verified. Residual normality was assessed through histogram and normal probability plot, homoscedasticity was evaluated using standardized residual plots, and multicollinearity was examined using tolerance and variance inflation factor values. Subsequently, outlier detection was conducted using standardized residuals, leverage values, and Cook's distance to identify influential cases. No extreme outliers were detected that warranted exclusion, and all data points were retained for analysis.

To test whether sex moderated the relationship between hip abduction strength and muscle activation, a moderation analysis was conducted using the PROCESS macro, model 1 (version 4.0), which is an open-source package used for moderation analysis [19]. Hip abduction strength (1-RM absolute and relative to body mass) was entered as the

independent variable in separate analysis, EMG amplitude (either GMed or TFL) as the dependent variable, and sex (coded as 1 = male, 2 = female) as the moderator. The interaction term (strength × sex) was computed automatically by PROCESS. Statistical significance was set at $p < 0.05$. The Bonferroni correction was used, adjusting the significance threshold by dividing the alpha level (0.05) by the number of main tests conducted (four). Thus, results were considered statistically significant at an adjusted alpha level of 0.0125. The proportion of explained variance ($R^2$) and changes in $R^2$ due to the interaction were reported. Conditional effects of strength on EMG amplitude were examined separately for each sex when the interaction was significant. All statistical analyses were performed using SPSS 22.0 software (SPSS Inc., Chicago, USA).

## Results

Eighteen males (age: 26.4 ± 3.9 years; body mass: 84.7 ± 10.9 kg; height: 1.76 ± 0.04 m; resistance training experience: 78% over 12 months; absolute 1RM: 18.7 ± 3.1 kg; relative 1RM: 0.22 ± 0.03 kg/kg) and 18 females (age: 26.1 ± 4.8 years; body mass: 60.1 ± 8.4 kg; height: 1.62 ± 0.09 m; resistance training experience: 78% over 12 months; 78% using hormonal contraception; absolute 1RM: 12.3 ± 2.2 kg; relative 1RM: 0.20 ± 0.04 kg/kg) were evaluated.

In the absolute 1RM, for GMed, the overall model was statistically significant [$F_{(3, 32)}$ = 11.10, $p < 0.001$, explaining approximately 51% of the variance in EMG amplitude ($R^2 = 0.51$). There was a significant interaction between hip strength and sex ($\beta = -1.86$, $t(32) = -3.45$, $p = 0.002$, 95% CI = −2.9; −0.76). Conditional analyses revealed a significant effect of hip strength on EMG amplitude for females ($\beta = -2.24$, $p < 0.001$, 95% CI = −3.1; −1.3), but not for males ($\beta = -0.37$, $p = 0.247$, 95% CI = −1; 0.27), suggesting that lower strength was associated with greater activation only among females (Fig 1).

For TFL, the overall model was not statistically significant [$F_{(3, 32)}$ = 0.19, $p = 0.89$], explaining only 2% of the variance in EMG amplitude ($R^2 = 0.02$). There was no significant interaction between hip strength and sex ($\beta = -0.16$, $t(32) = -0.15$, $p = 0.88$, 95% CI = −2.3; 1.99), nor a significant main effect of absolute hip strength ($\beta = 0.59$, $t(32) = 0.39$, $p = 0.70$, 95% CI [−2.48, 3.60]) or sex ($\beta = 5.11$, $t(32) = 0.32$, $p = 0.75$, 95% CI [−27.50, 37.80]) (Fig 1).

When the analysis was performed with relative 1RM, the result was similar. For GMed, the overall model was statistically significant ($p < 0.001$), explaining approximately 53.2% of the variance in EMG amplitude ($R^2 = 0.53$). There also was a significant interaction between hip strength and sex ($p = 0.011$). Conditional analyses revealed a significant effect of hip strength on EMG amplitude for females ($p < 0.001$), but not for males ($p = 0.40$). For TFL, the overall model was not statistically significant ($p = 0.99$), explaining only 0.1% of the variance in EMG amplitude ($R^2 = 0.01$). There was no significant interaction between hip strength and sex ($p = 0.88$), nor a significant main effect of absolute hip strength ($p = 0.89$) or sex ($p = 0.88$). Finally, after the Bonferroni correction, the interaction between hip strength and sex for GMed remained statistically significant.

## Discussion

Our study aimed to investigate whether maximal hip abduction strength (absolute and relative) is associated with hip abductor EMG amplitude during single-leg stance, considering sex as a moderator. We observed that lower hip abduction strength was associated with greater activation of GMed only among females, without similar effect on TFL.

This study was motivated by the two reasonable explanations existing in the literature regarding the possibility of activating the muscles more or less during single-leg stance: more activation as a compensatory mechanism for reduced torque production capacity or less activation due to the inability of generating neural drive [20]. Applying similar concepts, hip abductor weakness has been frequently attributed as the reason for differences in EMG amplitude of hip abductors (mainly GMed) between groups in tasks involving single-leg stance [4,6,8].

We found that 51–53.2% of the variance in GMed amplitude during sustained single-leg stance was explained by hip abduction strength, sex, and their interaction. Specifically, the interaction between sex and strength indicated that lower levels of strength were associated with greater GMed amplitude only in females, which may support the hypothesis of a

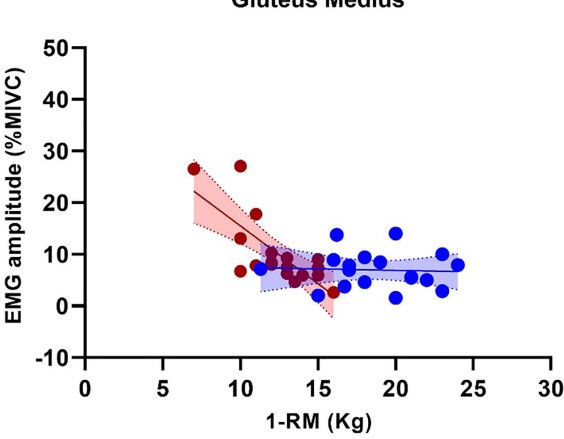

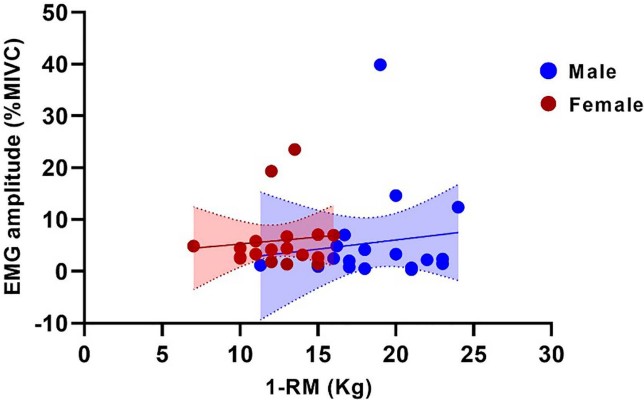

**Fig 1. Relationship between 1RM and EMG amplitude of GMed e TFL during single-leg stance.** Each point represents one participant. Solid red circles correspond to females, and blue circles represent males. Lines indicate the linear regression for each group, with shaded areas representing the 95% CI.

compensatory mechanism to maintain pelvic alignment [7,10]. Although we did not include kinematic measurements of the trunk or pelvis during our task, this mechanism is supported by a previous study showing that increased GMed activation was accompanying by reduced pelvic drop during walking while carrying a predetermined kettlebell weight in the contra-lateral hand [21]. Another possibility is that the lower tendon stiffness typically observed in females [22,23] may increase the demand for active muscle recruitment to stabilize the pelvis during single-leg stance. This hypothesis is explained by a compensated increase in muscle activation observed after a static-stretch intervention, causing a reduction in tendon stiff-ness [24], in order to improve the effectiveness of force generation and transmission [25]. However, in the absence of sim-ilar prior studies, this explanation remains speculative. Moreover, it's possible that there is a limit to how much force levels can optimize neuromuscular economy, which might explain the absence of an association in males, given their greater strength compared to females. Finally, no comparable effect was observed for the TFL, likely due to its less prominent role in pelvic stability during single-leg tasks [26].

Our study found significant association during single-leg stance, which, although part of different single-leg tasks (e.g., walking, single-leg squat, single-leg landing, step-down) [1], is an isometric task. Therefore, to what extent our findings

can be applied to more dynamic tasks remains unexplored. However, we hypothesize that the results may be similar for the following reasons. First, it is known that greater pelvic drop is associated with greater hip adduction during step-down tasks [27] and that lower GMed EMG activation was associated with higher hip adduction angle in single-leg tasks such as squatting, step-down, and landing [28]. Also, a meta-analysis found that lower dynamic hip abduction strength was moderately associated with greater hip adduction in single-leg tasks [29]. In all cited studies, the samples were predominantly female. Taken together, the established link between greater pelvic drop and increased hip adduction, along with the associations of reduced hip abductor strength and lower GMed activation with increased hip adduction across various dynamic single-leg tasks, supports the relevance of our isometric single-leg stance findings to more dynamic functional activities, particularly in female populations.

Although this compensatory increase in GMed activation may be effective in the short term, females with lower hip abduction strength may experience early fatigue, leading to altered lower limb movement patterns. This has important implications for sports that require single-leg control (e.g., running, soccer, basketball) and may help explain the higher incidence of certain injuries in females compared to males [3,4,14]. Therefore, strengthening the hip abductors may be particularly important in females to improve lower limb alignment during single-leg tasks.

Some important limitations need to be acknowledged: (i) we used the 1-RM test to assess the maximal force of hip abductors. Although this test is strongly associated with force measured by maximal isometric voluntary contractions [30], it is less commonly used than other types of strength assessments; (ii) the GMed is often functionally subdivided into three regions, which may limit the applicability of our results to tasks where the hip joint is in different positions (e.g., more flexed) or involves greater hip abductor moments (e.g., single-leg landing or cutting); (iii) our sample consisted of 36 participants for the moderation analysis. Therefore, we conducted a post hoc power analysis of the interaction effect using G*Power ($f^2 = 0.59$, $\alpha = 0.05$, $n = 36$), which indicated a statistical power greater than 0.95 to detect the interaction between hip strength and sex.

## Conclusion

Females with lower hip abduction strength demonstrate greater GMed activation during single-leg stance, suggesting a sex-specific compensatory strategy. No similar effect was observed for TFL. These findings highlight the importance of considering sex in neuromuscular assessments of pelvic stability.

## Supporting information

**S1 Data. Database.**
(PDF)

## Author contributions

**Conceptualization:** Rodrigo Rodrigues, Francesca Chaida Sonda, Michele Fernandes Frigotto, Pablo Gaviraghi.

**Data curation:** Rodrigo Rodrigues, Francesca Chaida Sonda, Michele Fernandes Frigotto, Pablo Gaviraghi, Talita Molinari, Rodrigo Rabello.

**Formal analysis:** Rodrigo Rodrigues, Francesca Chaida Sonda, Michele Fernandes Frigotto, Pablo Gaviraghi, Talita Molinari, Nicolas da Silva Pereira, Matheus Iglesias Marques, Rodrigo Freire Guimarães, Eduarda Bastos Cabral, Rodrigo Rabello.

**Methodology:** Rodrigo Rodrigues, Rodrigo Rabello.

**Project administration:** Rodrigo Rodrigues, Pablo Gaviraghi, Rodrigo Rabello.

**Supervision:** Rodrigo Rodrigues.

**Writing – original draft:** Rodrigo Rodrigues, Francesca Chaida Sonda, Michele Fernandes Frigotto, Pablo Gaviraghi, Talita Molinari, Nicolas da Silva Pereira, Matheus Iglesias Marques, Rodrigo Freire Guimarães, Eduarda Bastos Cabral, Rodrigo Rabello.

**Writing – review & editing:** Rodrigo Rodrigues, Francesca Chaida Sonda, Michele Fernandes Frigotto, Pablo Gaviraghi, Talita Molinari, Nicolas da Silva Pereira, Matheus Iglesias Marques, Rodrigo Freire Guimarães, Eduarda Bastos Cabral, Rodrigo Rabello.

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
