## [Decision Letter · Decision Letter 0]

14 Jul 2025

PONE-D-25-29454SEX AS A MODERATOR OF THE RELATIONSHIP BETWEEN HIP ABDUCTION STRENGTH AND MUSCLE ACTIVATION DURING SINGLE-LEG STANCEPLOS ONE

Dear Dr. Rabello,

Thank you for submitting your manuscript to PLOS ONE. After careful consideration, we feel that it has merit but does not fully meet PLOS ONE’s publication criteria as it currently stands. Therefore, we invite you to submit a revised version of the manuscript that addresses the points raised during the review process. In particular, please make sure to address Reviewer 2 comments to include more methodological details and how the findings can be generalized to clinical practice. 

We look forward to receiving your revised manuscript.

Kind regards,

Valentina Graci, PhD

Academic Editor

PLOS ONE

Journal Requirements:

Reviewers' comments:

Reviewer's Responses to Questions

**Comments to the Author**

1. Is the manuscript technically sound, and do the data support the conclusions?

Reviewer #1: No

Reviewer #2: Yes

2. Has the statistical analysis been performed appropriately and rigorously? 

Reviewer #1: No

Reviewer #2: Yes

3. Have the authors made all data underlying the findings in their manuscript fully available?

Reviewer #1: No

Reviewer #2: No

4. Is the manuscript presented in an intelligible fashion and written in standard English?

Reviewer #1: Yes

Reviewer #2: Yes

5. Review Comments to the Author

Reviewer #1: Dear Authors,

Thank you for the opportunity to review your manuscript. This study investigates an important and timely topic: whether sex moderates the relationship between hip abduction strength and muscle activation during a single-leg stance. The research question is well-motivated, and the finding of a sex-specific negative association in the gluteus medius is intriguing. The manuscript is generally well-written.

However, I have identified several major issues that need to be addressed before the manuscript can be considered for publication. My comments are detailed below.

Major Comments

1.The normalization of the TFL EMG data is a significant concern. The authors state that EMG amplitude was normalized to the MVIC of the GMed. As these are two different muscles with distinct functions and activation capacities, normalizing TFL activity to GMed's MVIC is methodologically inappropriate and likely invalidates all results reported for the TFL. The authors should either provide a strong justification for this unorthodox method or, ideally, re-normalize the TFL data using its own MVIC if that data is available. Otherwise, all analyses and conclusions regarding the TFL should be considered for removal from the manuscript.

2.The regression models do not appear to control for important potential confounders such as body mass and training history, which are critical confounders in strength analyses comparing males and females. I recommend including body size or composition covariates in the model or re-analyzing strength as a relative measure (e.g., normalized to body mass or lean mass) to ensure that the observed effects are not driven by body size alone.

3.The discussion repeatedly frames the increased GMed activation in weaker females as a compensatory strategy for pelvic stabilization. However, this interpretation assumes a functional outcome (i.e., improved pelvic control) that was not directly measured. Without kinematic data or stability-related performance outcomes, such as pelvic drop or trunk sway, this explanation remains speculative. Please consider revising the language to reflect that this is a hypothesis rather than a demonstrated mechanism, or alternatively, support the claim with additional evidence or references.

4.With only 36 participants and a sex-based interaction model, the power to detect interaction effects may be limited, especially given the potential data variance between groups. This raises concerns about Type II errors for nonsignificant findings (e.g., in males or for TFL) and potential overestimation of the observed effects in females. I suggest including a post-hoc power analysis for the interaction term, or at least discussing the limitations imposed by the sample size more explicitly in the Discussion section.

Minor Comments

1.For transparency and to assess estimate precision, I recommend reporting 95% confidence intervals (CIs) alongside β coefficients and p-values in all regression tables and/or main results text.

2.The manuscript does not report how failure to maintain single-leg stance was defined, or whether any trials were excluded due to poor task execution. This information is important for reproducibility. Please add a clear definition of task success/failure and any related exclusion criteria in the Methods section.

3.It is unclear whether participants performed the single-leg stance with eyes open or closed, which could affect balance strategy and neuromuscular activation. Please specify the visual condition used during testing and, if not standardized, discuss it as a limitation.

4.Although the interaction between sex and strength was not significant for TFL, it would be helpful to report or visualize sex-specific regression slopes, for comparison and transparency. If data permit, I suggest including a supplemental figure or text description.

5.The paper does not report whether regression assumptions were tested, such as residual normality, homoscedasticity, or multicollinearity. Please include a brief statement indicating that these assumptions were verified, or report diagnostics if available.

6.Since multiple regression models were used to test similar hypotheses for GMed and TFL, a correction for multiple testing (e.g., Bonferroni, Holm) may be warranted. If not applied, please justify this choice and address it as a potential limitation.

7.The 1RM estimation from submaximal trials is less common than MVIC in EMG studies and may introduce variability. While the authors cite evidence for its validity, a brief justification for selecting this method over MVIC would strengthen the Methods section.

8.The authors briefly mention lower tendon stiffness in females as a potential reason for increased GMed activation but do not provide enough detail. Expanding this part of the discussion would enhance the physiological plausibility of the interpretation.

9.Figure 1 provides a helpful visualization of the data. To further improve its clarity and impact, I suggest the authors add the regression lines for each group (males and females) to both plots. Including the 95% confidence bands for these lines would also be beneficial for interpreting the strength and precision of the observed associations.

10.In regression models with significant interaction terms, interpreting main effects independently can be misleading. Please consider rephrasing or de-emphasizing statements about the overall effect of strength across sexes, and focus interpretation within each subgroup.

11.The manuscript does not include a Data Availability Statement. The PLOS ONE policy requires such a statement to clarify for readers how the data can be accessed. Please add a formal statement detailing where the data is located or explaining any restrictions on its availability.

I look forward to seeing a revised version of this manuscript.

Reviewer #2: I appreciate the opportunity to review the manuscript entitled, “SEX AS A MODERATOR OF THE RELATIONSHIP BETWEEN HIP ABDUCTION STRENGTH

AND MUSCLE ACTIVATION DURING SINGLE-LEG STANCE” submitted to PLOS One. This study examined the impact of sex on hip abductor muscle function in the context of single leg stance as it relates to maximal muscle strength and muscle activation. The authors found that lower hip abduction strength was associated with greater activation of the glutes medius muscle in females but not males. The author argues this could represent a sex -specific compensatory strategy for this single leg stance control mechanism. Overall, this is a well written manuscript and in general the data supports the conclusion. I have a few questions and comments I would like the authors to address in terms of the methods and added discussion on the generalizability of the findings to clinical practice. I have provided section by section comments for the authors to consider and address.

Abstract: No edits

Introduction:

Overall, the introduction is well written and organized.

Methods:

Did the authors conduct an a priori power analysis to determine an adequate sample size for this study? If not, please justify the rationale for not doing one and add this to the manuscript.

Lines 108 -109 – Please mention this was a convenience sample.

Line 141 – Please provide the SENIAM guideline reference.

Lines 150-154 – Although the authors provide a reference with the detailed test description, I would add a short sentence or two that summarizes the methods of this estimated on rep max test. You must consider readers who are clinicians and do not have access to these important references for understanding the context of your study, especially since this method is a cornerstone of this investigation as it is how you measure muscle strength.

Lines 158 – I would briefly state that the PROCESS macro is an open-source package used for moderation analysis.

For example, I would just add: “…a moderation analysis was conduced using the PROCESS macro, model 1(version 4.0), which is an open-source package used for moderation analysis.”

Results:

Figure 1 – It would be helpful if the authors added trendlines indicating the relationship for each group, this would include showing the different slopes of the relationship.

Did the authors assess the data in figure 1 for outliers? This was not mentioned in the statistical analysis section. Linear models such as these are highly sensitive to outliers especially small samples.

Discussion:

The discussion does clearly report on the main findings and discusses them in the context of the other literature related to these findings. However, I would like to suggest that the authors consider their finding in the context of single limb control during a dynamic task like walking or single leg squat tasks. Do the authors think the relationship between would be sustained?

I think the authors need to discuss how generalizable these findings are to actual clinical practice. I would argue that in an athletic or highly active population that movement biomechanics of the participant play into this moderation based on dynamics of the system during the task. I would suggest adding a paragraph to discuss the generalizability of the findings to practice.

References: OK

Figure: See comment on adding linear regression lines to show different relationships among sex for each variable.

6. PLOS authors have the option to publish the peer review history of their article (what does this mean? ). If published, this will include your full peer review and any attached files.

**Do you want your identity to be public for this peer review?** For information about this choice, including consent withdrawal, please see our Privacy Policy .

Reviewer #1: No

Reviewer #2: No

---

## [Author Response · Author response to Decision Letter 1]

29 Jul 2025

Please find the detailed response attached.

---

## [Decision Letter · Decision Letter 1]

19 Aug 2025

SEX AS A MODERATOR OF THE RELATIONSHIP BETWEEN HIP ABDUCTION STRENGTH AND MUSCLE ACTIVATION DURING SINGLE-LEG STANCE

PONE-D-25-29454R1

Dear Dr. Rabello,

We’re pleased to inform you that your manuscript has been judged scientifically suitable for publication and will be formally accepted for publication once it meets all outstanding technical requirements.

Kind regards,

Valentina Graci, PhD

Academic Editor

PLOS ONE

Additional Editor Comments (optional):

Reviewers' comments:

Reviewer's Responses to Questions

**Comments to the Author**

1. If the authors have adequately addressed your comments raised in a previous round of review and you feel that this manuscript is now acceptable for publication, you may indicate that here to bypass the “Comments to the Author” section, enter your conflict of interest statement in the “Confidential to Editor” section, and submit your "Accept" recommendation.

Reviewer #1: All comments have been addressed

Reviewer #2: All comments have been addressed

2. Is the manuscript technically sound, and do the data support the conclusions?

Reviewer #1: Yes

Reviewer #2: Yes

3. Has the statistical analysis been performed appropriately and rigorously? 

Reviewer #1: Yes

Reviewer #2: Yes

4. Have the authors made all data underlying the findings in their manuscript fully available?

Reviewer #1: Yes

Reviewer #2: Yes

5. Is the manuscript presented in an intelligible fashion and written in standard English?

Reviewer #1: Yes

Reviewer #2: Yes

6. Review Comments to the Author

Reviewer #1: Dear Authors,

Thank you for your careful revisions and detailed responses to the comments from the previous round of review. I appreciate the effort you have put into improving the manuscript.

I have carefully reviewed the revised version and find that you have satisfactorily addressed all of my previous major and minor comments. In particular, I am pleased with the substantive changes you implemented, including correcting the TFL EMG normalization, re-analyzing for potential confounders, and adding a post-hoc power analysis. The expanded discussion, now supported by additional references, substantially enhances the physiological plausibility of your conclusions.

Overall, the manuscript is technically sound, the data support the conclusions, and it is clearly and coherently presented. I believe it is now ready for publication in *PLOS ONE*.

I commend the authors for their substantial efforts in strengthening the quality and clarity of the work.

Sincerely,

Reviewer #2: (No Response)

7. PLOS authors have the option to publish the peer review history of their article (what does this mean? ). If published, this will include your full peer review and any attached files.

**Do you want your identity to be public for this peer review?** For information about this choice, including consent withdrawal, please see our Privacy Policy .

Reviewer #1: No

Reviewer #2: No

---

## [Editor Report · Acceptance letter]

PONE-D-25-29454R1

PLOS ONE

Dear Dr. Rabello,

I'm pleased to inform you that your manuscript has been deemed suitable for publication in PLOS ONE. Congratulations! Your manuscript is now being handed over to our production team.

Kind regards,

on behalf of

Dr. Valentina Graci

Academic Editor

PLOS ONE